# Reinforced Target-driven Conversational Promotion

**Huy Quang Dao**[1,2], **Lizi Liao**[2], **Dung D. Le**[3], **Yuxiang Nie**[4]

[1]FPT Software AI Center, [2]Singapore Management University
[3]College of Engineering and Computer Science, VinUniversity
[4]Hong Kong University of Science and Technology
huydao98.uet@gmail.com, lzliao@smu.edu.sg
dung.ld@vinuni.edu.vn, ynieae@connect.ust.hk

## Abstract

The ability to proactively engage with users towards pitching products is highly desired for conversational assistants. However, existing conversational recommendation methods overemphasize on acquiring user preferences while ignore the strategic planning for nudging users towards accepting a designated item. Hence, these methods fail to promote specified items with engaging responses. In this work, we propose a **R**einforced **T**arget-driven **C**onversational **P**romotion (RTCP) framework for conversational promotion. Specifically, RTCP integrates short-term and long-term planning via a balanced gating mechanism. Inside which, the dialogue strategies are predicted via knowledge-integrated multi-head attention and guided via reinforcement learning rewards. RTCP then employs an action-guided prefix tuning method to generate relevant responses. Experimental results demonstrate that our model outperforms state-of-the-art models on both automatic metrics and human evaluation. Moreover, RTCP has a strong capability in quickly adapting to unseen scenarios just by updating prefix parameters without re-training the whole model. Code and data are here [1].

## 1 Introduction

In recent years, conversational recommender systems (CRSs) (Li et al., 2018; Kang et al., 2019; Zhou et al., 2020b; Ma et al., 2021; Zhou et al., 2021b; Li et al., 2022; Ren et al., 2022; Zou et al., 2022; Chu et al., 2023) have gained considerable attention from both academic researchers and industrial practitioners. Inside such systems, most CRS models will first infer user preferences via multi-turn conversations and then recommend a set of potential items when appropriate. The essential goal for such models is to solicit user interests accurately and then map such interests to some specific items in repository.

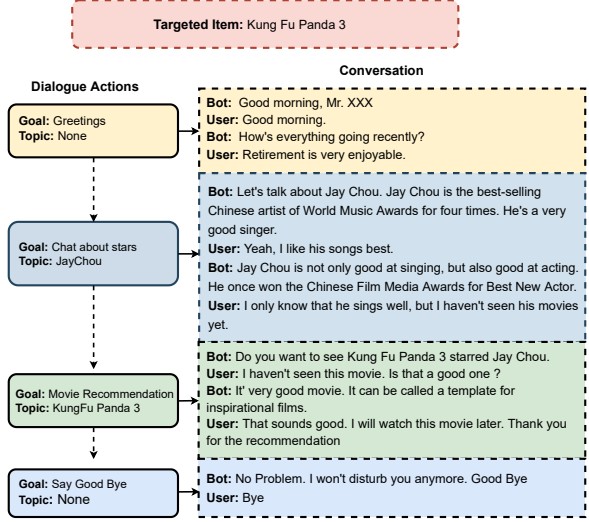

Figure 1: An illustration of the target-driven conversational promotion setting.

However, a more challenging scenario would be target-driven conversational promotion, where the system mimics a salesperson to nudge the users towards accepting a pre-given item. As illustrated in Figure 1, the bot tries to recommend a given target 'Kung Fu Panda 3' to the user via multiple turns of conversation. It gradually nudges the user via chatting about 'JayChou', on his well-known achievements in music, and then recommends his movie, which helps the user to get interested and accept the target with higher chance. Such a scenario is highly desired due to the potential for boosting sales revenue and reducing human costs but also calls for higher requirements on wisely promoting items and proactively engaging with users (Liao et al., 2023a,b; Deng et al., 2023a). Although existing CRS models (Chen et al., 2019; Zhou et al., 2020a; Lu et al., 2021; Zhou et al., 2021a; Liang et al., 2021; Zhou et al., 2022; Wang et al., 2022c) have achieved promising results, they tend to adopt a relatively *passive* scheme—mainly guessing user preferences via conversations, hence cannot be directly applied to the new proactive nudging setting.

Generally speaking, the new conversational pro-

---

[1]https://github.com/huyquangdao/RTCP

motion setting brings in two new obvious challenges that press for solutions. Firstly, as the virtual assistant needs to converse gradually to convince the user in accepting the target item, it is essential to produce an appropriate plan, e.g. in a sequence of action tuples, that leads to the target item while can also keep the user interested in the conversation. Secondly, once the plan is predicted, the other important requirement is to generate proper responses that are in line with the plan accurately. Also, it is likely that the model would encounter new conversation scenarios with unseen dialogue plans. The model should be able to adapt to such scenarios quickly and handle them properly, in order to avoid disappointing the users.

To address the aforementioned challenges, we propose a systematic framework–**R**einforced **T**arget-driven **C**onversational **P**romotion (RTCP) for the new setting. For planning, RTCP integrates long-term planning and short-term planning via a balanced gating mechanism. The former takes care of achieving the predefined target, while the latter looks into engaging the user and the smoothness of the conversation. Specifically, RTCP predicts action tuples in a plan via knowledge-integrated multi-head attention and is guided via estimated Reinforcement Learning rewards. For response generation, RTCP employs a novel action-guided prefix tuning method to regulate the response generation being in line with the plan. In particular, it pushes action plan factors into different prefix parameters, while leaves the main generation task to the pre-trained language model. This helps the model to learn and adapt better, especially when new action plans happen. To sum up, our contributions are threefold:

- We emphasize on a new target-driven conversational promotion setting where the agent nudges the user towards accepting a specified item via multiple turns of conversation.
- We propose a reinforced target-driven conversational promotion framework that balances between short-term and long-term planning, while further integrates action-driven prefix tuning to better guide the generation of relevant responses.
- Experiments show that RTCP outperforms the baselines on both automatic metrics and human evaluation. It also has a strong ability to quickly adapt to unseen conversation scenarios without re-training the whole model.

## 2 Related Work

### 2.1 Conversational Recommender Systems

The goal of CRS models is to offer personalized recommendations via interactive dialogues. One line of CRS methods (Lei et al., 2020a,b; Ren et al., 2021; Deng et al., 2021; Hu et al., 2022; Tu et al., 2022) mainly focuses on improving the performance of item recommendation, where they ask clarifying questions to gradually find an optimal candidate set. Therefore, the quality of generated responses is less emphasized as these works only leverage pre-defined response templates (Lei et al., 2020a,b; Ren et al., 2021) to interact with the users.

Another line of CRS research integrates recommendation modules into dialogue systems so that the item recommendation and the response generation objectives could be jointly optimized. Some efforts (Chen et al., 2019; Zhou et al., 2020a; Liao et al., 2020; Liang et al., 2021; Lu et al., 2021; Wang et al., 2022b; Zhou et al., 2022; Zhang et al., 2023) utilize different knowledge resources, such as knowledge graphs (Bizer et al., 2009; Wu et al., 2022) and user reviews to enrich extracted information from the conversations. However, such CRS systems lack appropriate conversational strategies to interact with users. Therefore, other recent methods (Liu et al., 2020, 2021; Hayati et al., 2020; Zhang et al., 2021) aim to develop goal-oriented CRS models that can converse with users by using different strategies (e.g chitchat, asking questions or recommendation). Liu et al. (2020) make the first attempt by introducing a goal-oriented CRS dataset called DuRecDial. State-of-the-art goal-driven CRS approaches (Wang et al., 2022a; Deng et al., 2023b) attempt to improve their short-term planning abilities with either target-side information or multi-task learning.

However, since these models mainly produce items according to the user's preference, it is nontrivial for these methods to transfer into a setting when a target item is given by the system beforehand. In this work, we utilize both short-term and long-term planning and combine these two strategies via a flexible balancing mechanism.

### 2.2 Controlled Response Generation

Recently, a surge of works (Zhong et al., 2021; Qin and Eisner, 2021; Ye et al., 2022b,a) focuses on leveraging the power of pre-trained language models for improving response generation. Among them, prefix-based methods aim to tune a set of

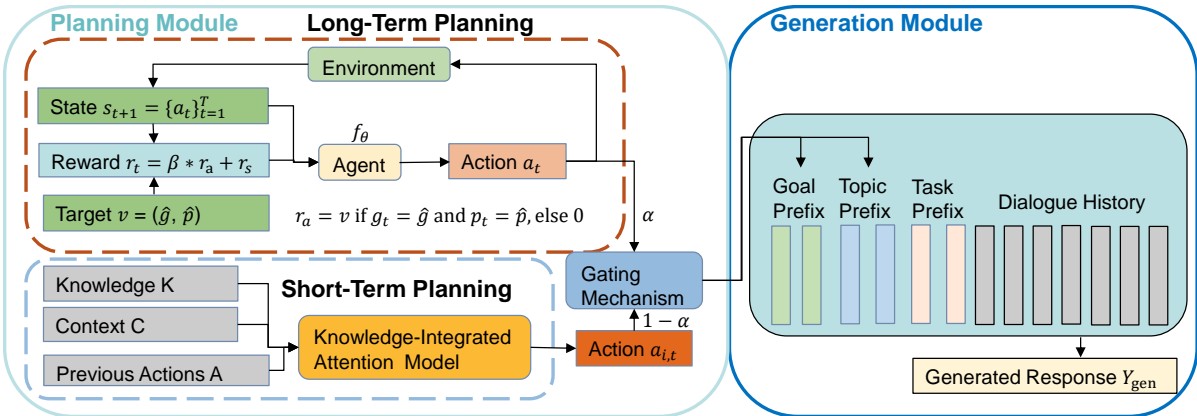

Figure 2: The architecture of the proposed RTCP model. The planning module predicts action tuples via balancing between short-term and long-term planning to keep user engaged and approach the target. With predicted action tuples, the generation module employs action-guided prefix tuning to generate relevant responses and realize fast adaptation to unseen scenarios.

trainable parameters, which are prepended to the input and control text generation (Li and Liang, 2021). Lester et al. (2021) propose a similar method, where they prepend the input sequence with special tokens and then they directly train the embeddings of these tokens. Such methods sidewalk the heavy burden of training separate large models for different settings. Inspired by these, we incorporate predicted strategies into separate prefix parameters and give different inputs to these prefixes for controlled response generation.

## 3 Preliminary

**Notations.** We denote by $\mathcal{G}$ the set of all goals (e.g. chit-chat, movie recommendation, etc), and denote by $\mathcal{P}$ the set of all topics (e.g The Conjuring, Jaychou, etc). At the $t$-th turn, we represent an action tuple $a_t$ by a pair of goal $g_t$ and topic $p_t$ (i.e $a_t = (g_t, p_t)$). Each conversation $D = (C, K, A)$ is a tuple of three elements namely the conversation content $C$, a set of all associated knowledge $K = \{k_j\}_{j=1}^{N_K}$ ($N_K$ is the size of the knowledge base) and a sequence of action tuples $A = \{a_t\}_{t=1}^{M}$ (M is the number of turns). The conversation context $C = \{(X_t, Y_t)\}_{t=1}^{M}$ is a set of all historical user utterances and corresponding responses. The goal of conversational promotion is to interact with the users to persuade them to accept a targeted item. We decompose the problem into 2 sub-tasks including (1) the dialogue strategy planning and (2) the response generation tasks.

**Dialogue Strategy Planning:** Formally, given a designated item $v \in \mathcal{P}$. We aim to produce a plan $\hat{A} = \{\hat{a}_1, \hat{a}_2, ..., \hat{a}_M\}$ which is a sequence of

action tuples to please the users and recommend the targeted item when appropriate.

**Response Generation:** For the $t$-th turn, given historical context $X_t$, knowledge base $\mathcal{K}$ and a predicted action tuple $\hat{a}_t$, we aim to generate a coherent response $\hat{Y}_t$ to the user.

## 4 Methodology

We show the overall framework in Figure 2. Our RTCP consists of two components namely planning in Section 4.1 and generation in Section 4.2.

### 4.1 Long Short-Term Strategic Planning

At each turn, the planning module needs to predict an action tuple $\hat{a}_t = (\hat{g}_t, \hat{p}_t)$. For target-driven recommendation setting, it is crucial to balance between two objectives: (1) engaging users with relevant topics; (2) recommending the target item while avoiding lengthy conversations. Correspondingly, we adopt short-term and long-term planning to achieve these two aforementioned objectives.

#### 4.1.1 Short-Term Planning

Similar to existing works (Zhang et al., 2021; Deng et al., 2023b), we consider short-term planning as the task of predicting the next action tuple.

**Knowledge-integrated Attention** To effectively leverage different kinds of inputs, we propose to utilize a Knowledge-integrated Attention block to gradually incorporate the previous action tuple sequence $A$, historical context $X$, and knowledge base $K$ in order to produce the final prediction (as

described as follows):

$$E_O^{(l)} = \text{MHA}(E_O^{(l-1)}, E_O^{(l-1)}, E_O^{(l-1)}),$$
$$E_O^{(l)} = \text{MHA}(E_O^{(l)}, E_A, E_A),$$
$$E_O^{(l)} = \text{MHA}(E_O^{(l)}, E_X, E_X),$$
$$E_O^{(l)} = \text{MHA}(E_O^{(l)}, E_K, E_K),$$
$$E_O^{(l)} = \text{FFN}(E_O^{(l)}),$$

where $E_O^{(l)}$ is the output of the $l$-th attention block while $E_A, E_X, E_K$ are representations of the action path, dialogue context, and knowledge base, which are computed by using separated BERT models (Devlin et al., 2019). MHA and FFN are the multi-head attention and feed-forward neural functions (Vaswani et al., 2017) respectively.

**Prediction via Short-Term Planning.** We utilize the representation of the [CLS] token of the last layer to simultaneously predict the next goal and topic. Formally, we compute probability distributions of goals and topics as follows:

$$\mathbf{P}_g = \text{Softmax}(W_g E_{O,[cls]}^{(L)} + b_g),$$
$$\mathbf{P}_p = \text{Softmax}(W_p E_{O,[cls]}^{(L)} + b_p),$$

where $\mathbf{P}_g, \mathbf{P}_p$ are probability distributions for the goals and topics respectively while $W_g, W_p, b_g, b_p$ are model's parameters. To train goal and topic planning, we minimize the negative log likelihood functions of the ground-truth goal $g^*$ and topic $p^*$:

$$L_g = -\sum_{i=1}^{N} \log \mathbf{P}_g(g_i^*|X_i, K_i, A_i, \Theta_g),$$
$$L_p = -\sum_{i=1}^{N} \log \mathbf{P}_p(p_i^*|X_i, K_i, A_i, \Theta_p),$$

where N is the total number of training examples and $\Theta_g, \Theta_p$ are parameters associated with the goal and topic prediction task respectively.

### 4.1.2 Long-Term Planning

In this work, we adopt Reinforcement Learning (RL) to enhance the long-term planning ability of our system. The goal is to encourage the system to focus more on recommending the targeted item while avoiding long-lasting conversations.

**A MDP Viewpoint of Conversational Planning.** We re-formalize conversational planning as a Markov Decision Process (MDP) which includes

a sequence of states, actions, and rewards. Formally, a MDP consists of a tuple of five elements $\mathcal{M} = (\mathcal{S}, \mathcal{A}, \mathcal{P}, \mathcal{R}, \gamma)$. At the $t-$th turn, we define the state $s_t \in \mathcal{S}$ as the sequence of previous $t-1$ action tuples in chronological order (i.e $s_t = \{a_1, a_2, ..., a_{t-1}\}$). The action space $\mathcal{A}$ consists of all available action tuples (i.e $\mathcal{A} = \{a_j = (g_j, p_j)|g_j \in \mathcal{G}, p_j \in \mathcal{P}\}$). For state transition, after employing the action tuple $a_t$, we append the action tuple $a_t$ to the end of the current state $s_t$ to construct the next state $s_{t+1}$. Finally, $\gamma \in [0, 1]$ is the discount factor.

**Reward $\mathcal{R}$.** To train the RL agent, we define the following reward function:

$$r_t = \beta * r_a + r_s,$$

where $r_t$ is the final reward and $\beta$ is a hyperparameter while we refer to $r_a, r_s$ as target and intermediate rewards respectively. On the one hand, we use the former to encourage the agent when it successfully recommends the targeted item. The target reward is computed as follows:

$$r_a = \begin{cases} v, & \text{if } g_t = \hat{g}, p_t = \hat{s}, t_1 \leq t \leq t_2, \\ 0, & \text{otherwise}, \end{cases}$$

where $v$ is a predefined value, $\hat{g}, \hat{s}$ are the targeted goal and item of the current conversation respectively. $t_1, t_2$ are two hyper-parameters utilized to determine whether the system should recommend the targeted item. On the other hand, we adopt the intermediate reward to preserve the relevance of topical transitions during the conversations and compute it by applying a binary classification model on the state $s_{t+1}$ as follows:

$$r_s = \sigma(W_s h_{s_{t+1}} + b_s),$$

where $h_{s_{t+1}}$ is the latent representation of the state $s_{t+1}$, $\sigma$ is the Sigmoid activation function and $W_s, b_s$ are parameters. To compute $h_{s_{t+1}}$, we feed the next state $s_{t+1}$ through a BERT model and utilize the output corresponding to the [CLS] token. Additionally, to train the binary classification model, we consider sequences of strategies from the training set as positive instances and randomly replace some elements in the positive instances to produce negative ones.

**Prediction via Long-Term Planning.** At each turn, we compute a policy $\pi(a_t|s_t, \theta)$ which is a

probability distribution over all available action tuples by using the following formulation:

$$\pi(a_t|s_t, \theta) = f_\theta(s_t),$$

where function $f_\theta$ is parameterized by a neural network and $\theta$ are its corresponding parameters. Finally, following (Zhao et al., 2017), we utilize actor-critic framework to train our RL agent.

### 4.1.3 Long Short-Term Strategic Balancing

Based on the conversation situation, the system will need to balance short-term and long-term planning for better performance. Therefore, we propose an adaptive gating mechanism that allows the system to flexibly combine the two aforementioned abilities. Specifically, during the inference step, we compute a weighted combination of two probability distributions. Formally, the final distribution on action tuples is computed as follows:

$$\mathbf{P}(a_t) = (1 - \alpha) * \mathbf{P}_{short}(a_t) + \alpha * \mathbf{P}_{long}(a_t)$$
$$= (1 - \alpha) * \mathbf{P}_g(g_t) * \mathbf{P}_p(p_t) + \alpha * \pi(a_t),$$

where $\alpha$ is a hyper-parameter that can be flexibly chosen by using cross-validation. $\mathbf{P}_{short}(a_t), \mathbf{P}_{long}(a_t)$ are probability distributions computed by the short-term and long-term parts respectively. Here, we assume that goal and topic are two independent random variables. Therefore, the short-term distribution $\mathbf{P}_{short}(a_t)$ can be computed by taking the product of the two factorial distributions. By varying $\alpha$, the planning strategy of the system also changes accordingly. A larger value of $\alpha$ might encourage the system to quickly suggest the targeted item to the user. In contrast, by using a smaller $\alpha$, the system focuses more on maintaining the relevance of topical transitions during the conversations. Consequently, choosing suitable values for $\alpha$ is not trivial and largely depends on the situation at hand.

### 4.2 Action-Guided Response Generation

Given a predicted action tuple $\hat{a}_t$ that consists of a goal $\hat{g}_t$ and a topic $\hat{p}_t$ from the planning part, the language generation model is expected to produce appropriate responses accordingly.

**Action-guided Prefix Tuning.** Inspired by methods from controlled generation, we enhance the standard prefix tuning (Li and Liang, 2021) with action-driven contents to further calibrate the generation process. Specifically, to train the language

generation model, we prepend learnable continuous prompts representing the predicted goal and topic to the input sequence. Formally, given a specific action tuple $\hat{a}_t = (\hat{g}_t, \hat{p}_t)$, we construct the input sequence $\mathcal{I}_t$ for the $i$-th training example as follows.

$$\mathcal{I}_t = [\mathbf{G}_t, \mathbf{P}_t, \mathbf{T}_{gen}, X_t],$$

where $X_t$ is the historical context while $\mathbf{G}_t, \mathbf{P}_t, \mathbf{T}_{gen}$ are corresponding continuous prompt tokens representing the goal, topic and the generation task respectively. Noticeably, in contrast to these existing prompt tuning methods (Li and Liang, 2021), our proposed approach is data-driven which means the model can adapt better to different individual training examples.

**Parameter Optimization.** We fix the parameters of pretrained GPT2 model and only learn the additional latent prompts. That means only the goal prompt $\mathbf{G}_t$, topic prompt $\mathbf{P}_t$, and task-specific tokens $\mathbf{T}_{gen}$ are updated during the training process. To learn the response generation task, we optimize the following loss function:

$$L_{gen} = -\sum_{t=1}^{N} \sum_{j=1}^{L} \log \mathbf{P}_{gen}(y_{t,j}^*|y_{t,<j}|\mathbf{G}_t,$$
$$\mathbf{P}_t, \mathbf{T}_{gen}, X_t, \Theta_{plm}),$$

where $N$ is the total number of training examples, $L$ is the length of the output sequence and $\Theta_{plm}$ are fixed parameters of the pretrained language model.

## 5 Experiments

### 5.1 Experimental Setup

**Dataset.** In this work, we utilize **DuRecDial** (Liu et al., 2021) and **INSPIRED** (Hayati et al., 2020) to conduct our experiments. Both of them are originally designed for goal-driven CRS setting. It is worth noticing that for **INSPIRED**, we only conduct the goal planning due to lacking annotations of topics. We show the detailed statistics of these two datasets in the Appendix A.2. In this work, we regard the item which the user will accept at the end of each conversation as the targeted item.

**Baselines.** In this work, we compare RTCP with several representative baseline methods including general pre-trained language models( **GPT2** (Radford et al., 2019), **DialoGPT** (Zhang et al., 2020), **BART** (Lewis et al., 2020) ) and state-of-the-art

goal-driven CRS methods (**KERS** (Zhang et al., 2021), **TCP** (Wang et al., 2022a) and **UNIMIND** (Deng et al., 2023b)). Moreover, we also report the results of variants of RTCP including RTCP without knowledge-integrated attention (**RTCP-KIA**), RTCP without goal-topic prompts (**RTCP-GP**) and RTCP without task-specific prompt (**RTCP-T**$_{gen}$). We describe the details of baselines in Appendix A.3.

**Implementation Details**  We use Pytorch framework [2] to implement our conversational promotion framework and train the framework on 1GPU NVIDIA A100 40G card. We run the model 4 times with different random seeds and compute the averaged results. In this work, we use the GPT2-base (114M) and Bert-base (114M) as our backbone models, We set the dimension of hidden vectors to 768. For the knowledge-integrated attention model, we set the number of layers to 12 (4 for INSPIRED), and the number of attention heads to 8. To train our RL module, we set the target reward and weighted parameter $\gamma$ to 3.0 and 1.0 respectively. To balance between short-term and long-term planning, we perform cross-validation and empirically set the value of $\alpha$ to 0.6 and 0.9 for DuRecDial and INSPIRED respectively. We use a learning rate of 1e-5 to train the actor network while the learning rate of the critic is set to 5e-5. For the response generation part, we use 50 soft tokens for training the task-specific prefix and 2 for both the goal and topic prefixes respectively. Finally, we train the generation model with a learning rate of 5e-5 with 5 epochs till converge.

**Evaluation Metrics.**  We evaluate the models on both dialogue planning as well as response generation aspects. In particular, we utilize accuracy (**Acc**) and joint accuracy (**Joint Acc**) metrics for short-term planning. For the target accomplishment, we report the dialogue-level success rate (**SR**), success rate at $k$-th turn (**SR@k**) (Lei et al., 2020a,b) and the averaged number of conversation turns (**Avg. Turns**) needed to successfully recommend the targeted item. For response generation, we utilize both automatic and human evaluation. For automatic evaluation, we use perplexity (**PPL**), word-level F1 (**F1**), **BLEU-N** (N=1,2) (Papineni et al., 2002), **Dist-N** (N =1,2) (Li et al., 2016) and Knowledge F1 (**Know. F1**)(Wang et al., 2022a; Zhang et al., 2021). For human study, we randomly

[2] https://pytorch.org/

| Model | DuRecDial | | | INSPIRED |
| | Goal Prediction Acc. (%) | Topic Prediction Acc. (%) | Joint Acc. Acc. (%) | Goal Prediction Acc (%) |
|---|---|---|---|---|
| KERS | 96.10 | 78.41 | 77.83 | 19.64 |
| BERT | 97.81 | 88.03 | 89.61 | 27.21 |
| TCP | 97.88 | 93.00 | 92.75 | 21.12 |
| UNIMIND | 97.26 | 92.67 | 92.37 | 22.93 |
| RTCP | **98.41** | **95.27** | **95.12** | **30.57** |
| RTCP - KIA | 98.28 | 93.08 | 92.83 | 24.17 |

Table 1: Results on the short-term planning, demonstrated via goal and topic prediction (t-test, $p < 0.05$).

| Model | DuRecDial (t = 0.6) | | | INSPIRED (t = 0.9) | |
| | SR@1 | SR | Avg. Turns($\downarrow$) | SR | Avg. Turns($\downarrow$) |
|---|---|---|---|---|---|
| DialoGPT | 17.06 | 53.96 | 4.61 | 7.46 | 17.08 |
| GPT2 | 14.77 | 67.22 | 4.26 | 3.33 | 17.53 |
| BART | **22.17** | 67.67 | 3.96 | 2.61 | 17.75 |
| KERS | 1.65 | 19.72 | 6.33 | 1.33 | 17.75 |
| TCP | 7.45 | 75.43 | 4.26 | 8.33 | 17.23 |
| UNIMIND | 8.13 | 86.89 | 3.80 | 10.04 | 17.21 |
| RTCP ($\alpha = t$) | 9.84 | **87.49** | **3.74** | **11.67** | **16.82** |
| RTCP ($\alpha = 0.0$) | 9.21 | 86.78 | 3.80 | 9.16 | 16.91 |
| RTCP ($\alpha = 0.0$) - GP | 8.91 | 87.44 | 3.77 | 7.49 | 17.16 |

Table 2: Results on the long-term planning, demonstrated via the target achievement (t-test, $p < 0.05$).

sample 20 dialogues and invite two annotators to score those dialogues. We report the results in both turn-level and dialogue-level. For turn-level, we concern three aspects, **Fluency**, **Proactivity** and **Informativeness**, while for dialogue-level we compare user **Satisfaction** and conversation **Cohenrency**. The range of scores is from 1 to 3. We measure the inter-annotator agreement by Fleiss' Kappa (McHugh, 2012).

## 5.2  Main Results

### 5.2.1  Results on Short-term Planning

Table 1 shows the empirical results of the goal and topic prediction tasks. Overall, pre-trained models such as BERT and TCP perform better than KERS. This is because such models are pre-trained on massive amounts of text, hence could produce more meaningful representations for downstream tasks. Second, we observe that our proposed method RTCP significantly outperforms all baseline methods across all metrics, which indicates the superior short-term planning ability of our model compared to others. The superior performance of our proposed RTCP mainly comes from the knowledge-integrated attention mechanism. This is supported by the substantial drop in the performance for the variant RTCP-KIA, in which we removed the knowledge-integrated attention.

| Model | DuRecDial (t = 0.6) | | | | | | | INSPIRED (t = 0.9) | | | | | | |
|---|---|---|---|---|---|---|---|---|---|---|---|---|---|---|
| | PPL ($\downarrow$) | F1 | BLEU-1 | BLEU-2 | DIST-1 | DIST-2 | Know. F1 | PPL ($\downarrow$) | F1 | BLEU-1 | BLEU-2 | DIST-1 | DIST-2 | Know. F1 |
| BART | 6.08 | 28.17 | 0.418 | 0.261 | 0.034 | 0.101 | 41.06 | 19.47 | 10.21 | 0.292 | 0.112 | 0.088 | 0.198 | 10.16 |
| DialoGPT | 4.59 | 37.11 | 0.452 | 0.314 | 0.030 | 0.095 | 42.20 | 16.69 | 14.58 | 0.338 | 0.142 | 0.050 | 0.117 | 12.42 |
| GPT2 | 4.04 | 39.24 | 0.495 | 0.349 | 0.035 | **0.111** | 48.19 | 15.72 | 13.71 | 0.356 | 0.153 | 0.085 | 0.218 | 8.41 |
| KERS | 6.69 | 35.41 | 0.473 | 0.278 | 0.009 | 0.027 | 33.19 | 24.81 | 10.54 | 0.228 | 0.102 | 0.013 | 0.030 | 1.23 |
| TCP | 4.15 | 41.11 | 0.507 | 0.362 | 0.035 | 0.107 | 60.81 | 17.11 | 13.50 | 0.352 | 0.149 | 0.087 | 0.227 | 7.38 |
| UNIMIND | 4.09 | 42.51 | 0.530 | 0.381 | 0.032 | 0.102 | 68.66 | 15.38 | 15.69 | 0.356 | 0.159 | **0.125** | **0.295** | 11.85 |
| RTCP ($\alpha = t$) | 4.36 | 42.27 | 0.524 | 0.376 | 0.034 | 0.104 | 63.13 | 19.72 | 14.27 | 0.351 | 0.153 | 0.085 | 0.211 | 14.24 |
| RTCP ($\alpha = 0.0$) | **3.69** | **45.39** | **0.542** | **0.402** | **0.036** | 0.109 | **70.35** | 18.64 | 15.39 | **0.371** | **0.168** | 0.091 | 0.211 | 10.21 |
| RTCP ($\alpha = 0.0$) - GP | 3.89 | 44.66 | 0.529 | 0.391 | 0.034 | 0.102 | 68.04 | **13.48** | **17.31** | 0.355 | 0.167 | 0.077 | 0.187 | **17.16** |
| RTCP ($\alpha = 0.0$) - Tgen | 5.10 | 36.38 | 0.482 | 0.326 | 0.031 | 0.104 | 46.86 | 33.78 | 13.25 | 0.346 | 0.152 | 0.063 | 0.154 | 4.41 |

Table 3: Automatic evaluation results on the response generation task (t-test, $p < 0.05$). We empirically set $\alpha$ to 0.6 and 0.9 for DuRecDial and INSPIRED respectively.

| Model | DuRecDial | | | | | | INSPIRED | | | | | |
|---|---|---|---|---|---|---|---|---|---|---|---|---|
| | Turn-level results | | | Dialog-level results | | | Turn-level results | | | Dialog-level results | | |
| | Fluency | Infor. | Proactivity | Satisfaction | Coherency | Kappa | Fluency | Infor. | Proactivity | Satisfaction | Coherency | Kappa |
| GPT2 | 2.935 | 2.194 | 1.623 | 1.95 | 1.75 | 0.72 | 2.921 | 1.242 | 1.079 | 1.10 | 1.10 | 0.84 |
| KERS | 2.577 | 1.759 | 1.376 | 1.55 | 1.45 | 0.75 | 2.881 | 1.079 | 1.019 | 1.00 | 1.00 | 0.88 |
| TCP | 2.967 | 2.227 | 1.649 | 2.25 | 2.05 | 0.80 | 2.895 | 1.227 | 1.123 | 1.20 | 1.10 | 0.83 |
| UNIMIND | 2.971 | 2.236 | 1.659 | 2.32 | 2.22 | 0.79 | 2.932 | 1.504 | 1.148 | 1.50 | 1.20 | 0.87 |
| RTCP | **2.981** | **2.246** | **1.676** | **2.55** | **2.45** | 0.74 | **2.971** | **1.752** | **1.237** | **1.90** | **1.30** | 0.86 |

Table 4: Human evaluation results on the generated responses (**Infor.** stands for informativeness).

### 5.2.2 Results on Target-achievement Task

Table 2 shows the performance comparison on the target-achievement aspect. First, on DuRecDial, relatively general models such as DialoGPT, GPT2, and BART achieve rather good results on SR@1. The reason is these models tend to recommend the target item at the beginning state of the conversation since the target item is a part of their input sequences. However, such behavior is less preferred in terms of user engagement and satisfaction. On dialogue-level success rate (SR), target-driven methods such as TCP and UNIMIND perform significantly better than the aforementioned pretrained language models. Noticeably, our RTCP consistently outperforms all baseline methods and achieves state-of-the-art results. Moreover, compared to other baseline approaches, the proposed RTCP manages to complete the task in relatively shorter conversations. This can be attributed to RTCP's long-term planning part which is effectively guided by reinforcement learning rewards.

### 5.2.3 Automatic Evaluation on Responses

Table 3 shows the performance comparison on the response generation task based on automatic metrics. First, target-driven methods such as TCP and UNIMIND perform better than pre-trained language models (i.e. BART, GPT2, and DialogGPT). Thanks to their short-term planning part, such CRS methods could produce interactive plans

to effectively guide the response generation task. Second, our RTCP variants significantly outperform baseline methods on several metrics (6 out of 7 metrics on DuRecDial and 5 of 7 metrics on INSPIRED). This mainly comes from: (1): RTCP could produce more accurate conversational plans as demonstrated in Section 5.2.1. (2) RTCP utilizes action-guided prompt tuning which parameterizes dialogue strategies with learnable parameters and adjusts them to better calibrate the generation process. This is further demonstrated by removing the goal-topic prompts (RTCP ($\alpha = 0$ - GP) results in a considerable drop in the results. Similarly, we also experience a performance degradation when removing the task-specific prompt $\mathbf{T}_{gen}$, which indicates the general task-specific tokens are indeed important for the generation task. Finally, we show the performance of RTCP with different values of $\alpha$. RTCP ($\alpha = 0$) (only short-term planning) achieves the best results. This is as expected since this evaluation scheme is carried out turn-by-turn and hence favors those methods that directly optimize turn-level objectives.

### 5.2.4 Human Evaluation on Responses

Table 4 shows the results of human evaluation. Overall, compared to other baseline methods, our RTCP framework achieves better performance on both turn-level and dialogue-level assessments. Interestingly, in the dialogue level, despite achieving a high SR@1 as shown in Section 5.2.1, the GPT2

baseline still performs worse than TCP and TCP on the satisfaction and coherency metrics. This indicates that the annotators are not quite satisfied with responses generated by the GPT2 model. Such an interesting insight further demonstrates our observation that pre-trained language models (GPT2, BART, and DialoGPT) tend to recommend the targeted item at the early stage of the dialogues, which is not preferred in the conversational promotion setting. Unsurprisingly, our RTCP presents the best results on dialogue-level metrics. This could be attributed to our planning strategy, which could produce appropriate dialogue strategies to manage the conversations.

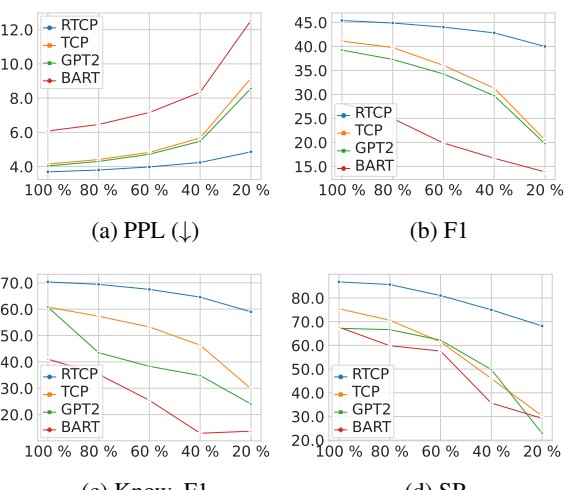

Figure 4: Performance comparison on DuRecDial w.r.t different ratio of the original training data.

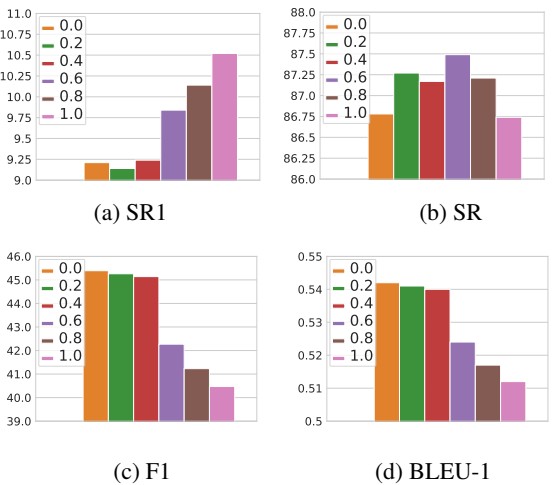

Figure 3: Performance of RTCP on DuRecDial with different values of balancing parameter $\alpha$.

## 5.3 Detailed Analyses

### 5.3.1 Effectiveness of Balancing Parameter $\alpha$

Figure 3 shows the results on SR@1, SR, F1 and BLEU-1 metrics of our RTCP model with different values of balancing hyper-parameter $\alpha$. First, we can see that as we increase the value of $\alpha$, the SR@1 and SR results gradually increase. This is reasonable since a large value of $\alpha$ means we focus more on recommending the target item. However, it also indicates that over-emphasizing long-term planning would also hurt the model's performance (as SR decreases). Moreover, we notice that for generation metrics such as F1 and BLEU-1, the performance result decreases as $\alpha$ increases. The main reason is that the F1 and BLEU-1 metrics work in turn level. When long-term planning contributes more, it is very likely that the model would choose new plans which are significantly different from ground-truth plans.

### 5.3.2 Effectiveness of Training Data Size

Figure 4 illustrates the performance of BART, GPT2, TCP, and the proposed RTCP on different training data sizes. Overall, we can see that RTCP consistently outperforms other baseline approaches in all metrics across all different training sizes. It indicates that RTCP learns faster and more effectively than other baselines. This might be due to the fact that RTCP only tunes prefix parameters for generation. The backbone generation model inside RTCP is a pre-trained GPT2 model which is fixed and will not be updated. For baselines such as BART, GPT2, and TCP, they have a large number of tunable parameters, which requires more training data to approach a good local optima position. Hence, when the ratio of training data decreases, these methods' performance drops relatively faster.

### 5.3.3 Fast Adaptation to Unseen Plans

We further conduct an experiment to demonstrate the adaptability of our RTCP model to unseen scenarios. First, from the set of all available topics, we randomly select a subset as unseen topics which is approximately 20% of all topics. Then we divide the original training set into two parts: *train_seen* and *train_unseen*, where the *train_unseen* set contains all the unseen topic samples. We do the same for the original testing set and obtain *test_seen* and *test_unseen*. Then we train the proposed RTCP and the comparing TCP model on *train_seen*. For the *Without Training* setting, we directly apply trained RTCP and TCP models on the *test_unseen* for results. For the *With Training setting*, we further fine-tune RTCP and TCP models on *train_unseen*

and test on the *test_unseen*.

| Model | | F1 | BLEU-1 | BLEU-2 | Know. F1 |
|---|---|---|---|---|---|
| Without Training | TCP | 36.15 | 0.419 | 0.298 | 55.04 |
| | RTCP | **43.51** | **0.532** | **0.381** | **69.38** |
| | RTCP - GP | 43.24 | 0.514 | 0.371 | 66.07 |
| With Training | TCP | 42.27 | 0.500 | 0.362 | 56.07 |
| | RTCP | **47.54** | **0.568** | **0.425** | **74.31** |
| | RTCP - GP | 45.72 | 0.541 | 0.401 | 71.71 |

Table 5: Performance comparison on unseen dialogue examples (results on DuRecDial) (t-test, $p < 0.05$).

The results are shown in Table 5. We observe that even when the models have not seen such scenarios (without training), RTCP manages to perform better than TCP by large margins across several metrics. When get exposed to new training cases and get fine-tuned, both TCP and RTCP obtain better performance results. However, there are two important things to note: (1) In order to fine-tune the model for such new scenarios, TCP needs to update the whole model while RTCP only needs to tune the prefix parameters, and the backbone GPT2 part remains unchanged. This is of much lower cost; hence RTCP is much more efficient as compared to other fine-tuning based approaches. (2) In the *With Training* setting, removing goal and topic prefixes results in a considerable drop in the performance, which indicates the impacts of these action-driven parameters to our RTCP when it is required to adapt to new plans.

## 6 Conclusion

In this work, we focused on a promising yet under-explored setting called target-driven conversational promotion. We proposed a novel RTCP framework to produce dialogue action tuples which direct the conversation to a targeted item while also please the user with interesting and relevant topics. It manages to properly generate responses that are closely in-line with the plan while also can adapt quickly to new scenarios. We conducted extensive experiments to compare with a bunch of representative baselines over a rich set of metrics. Both automatic evaluation and human evaluation results showed that RTCP significantly outperformed state-of-the-art approaches on many important aspects.

## Limitations

We discuss the limitations from the following perspectives: (1) **Difficulty level of promotion datasets.** In existing CRS datasets (DuRecDial, INSPIRED, etc.), the users are likely to accept items mentioned at the end of conversations, which eventually limits the difficulty of the new nudging setting. To make the task more challenging, it might be better to construct new promotion-oriented datasets where the systems are given "hard target items" that are less likely to be accepted by the users, and the systems need several rounds of promotions to convince the users. (2) **Usage of LLMs.** Recently, large language models (LLMs) such as ChatGPT or LLaMA have exhibited their out-standing performance on generation. We plan to further explore how to make use of such powerful LLM models on the target-driven promotion setting. (3) **Reliance on dialogue actions.** Existing goal-driven CRS models (KERS, UNIMIND) and our model heavily rely on a pre-defined set of dialogue actions, which inherently restricts their usages to real-world scenarios. Improving action-free conversational models will be a more potential solution to alleviate such reliance.

## Acknowledgements

This research was funded by Vingroup Innovation Foundation (VINIF) under project code VINIF.2022.DA00087 and the Singapore Ministry of Education (MOE) Tier 1 Academic Research Fund No. 21-SIS-SMU-038, MSS22C003.

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

## A Appendix

### A.1 Additional Comparison with Large Language Models (LLMs)

Recently, LLMs such as GPT3 (Brown et al., 2020), ChatGPT (Liu et al., 2023) and Llama (Touvron et al., 2023) have shown out-standing performance on a variety of NLP tasks. Hence, in this work, we also report the performance comparison between our RTCP model and these LLM-based target-driven recommender systems. In particular, we adopt ChatGPT and leverage zero-shot prompting to produce responses using the LLM model. We construct the input prompt by prepending the target item to the current dialogue context and ask the model to generate the corresponding response. As can be seen in Table 6, We can observe that RTCP significantly achieves better performance on several metrics including BLEU-N (N=1,2), F1 and Know. F1. This is reasonable since RTCP is optimized to mimic responses in the corpus. However, ChatGPT performs better than our RTCP on DIST-N metrics (N=1,2). This can be attributed to the fact that ChatGPT have been pre-trained on a massive amount of texts. Therefore, it could generate more diverse responses. For target-achievement aspects, we can observe that our RTCP outperforms ChatGPT on all metrics. This can be attributed to the planning module of RTCP which could produce appropriate plans to direct the conversations towards the target item, while ChatGPT tends to passively answer user queries. This can be seen in Table 7 which shows the results of human evaluation with RTCP and ChatGPT on DuRecDial. Athough ChatGPT is trained with large corpus and with huge amount of parameters, our model still manages to outperform it in Proactiveness.

### A.2 Detailed Statistics of Benchmark Datasets

Detailed statistics of datasets are described in Table 8. In particular, the **DuRecDial 2.0** dataset (Liu et al., 2021) consists of 16.5K English-Chinese parallel dialogues and approximately 55K natural language utterances belong to seven different domains such as Movie/POI/Food/Music/Weather, etc. In this work, we utilize the English version of the dataset. We adopt the default data split 6.5 : 1 : 2.5 for training/development/test respectively. On the other hand, **INSPIRED** dataset (Hayati et al., 2020) consists of 1001 conversations (35K utterances) that belong to the movie recommendation domain. The dataset supports sociable conversation recom-

mendation setting and there are 19 different social strategies in INSPIRED. In our setting, we regard these social strategies as goals and attempt to predict them at each turn. Finally, we use the default data splits of INSPIRED to conduct our experiments.

### A.3 Additional Details of Baseline Methods

In this work, we compare our RTCP framework against several representative baselines including:

- **BERT** (Devlin et al., 2019) is a widely used baseline model for text classification tasks. We adapt it to show the performance comparisons on goal and topic prediction.
- **GPT2** (Radford et al., 2019) is a basic but strong text generation baseline which gains from large pretrained language modeling.
- **BART** (Lewis et al., 2020) is a more recent denoising autoencoder pretrained model for language generation. It can be seen as generalizing BERT, GPT, and many other more recent pretraining schemes.
- **DialoGPT** (Zhang et al., 2020) is a dialogue generative pre-trained GPT model trained on large-scale conversation-like exchanges from Reddit, which helps to generate more relevant and context-consistent responses.
- **KERS** (Zhang et al., 2021) is a knowledge-enhanced multi-subgoal driven recommender which predicts a sequence of subgoals and use them to guide the dialog model to select knowledge from a sub-set of existing knowledge graph.
- **TCP** (Wang et al., 2022a) is a target-driven recommender. It predicts a sequence of goals and topics in short-term planning manner which further guide the response generation.
- **UNIMIND** (Deng et al., 2023b) is the state-of-the-art goal-driven CRS model which leverages a multi-task learning and a prompt-based approach to unify sub-tasks of multi-goal CRS setting.

### A.4 Instructions for Human Evaluation

Given generated conversations, we ask the annotators to evaluate those dialogues in both turn-level and dialogue-level aspects. For turn-level results, we report three metrics namely Fluency, Informativeness, and Proactivity. On the dialogue level, we report Satisfaction and Coherency respectively. The rating scores must be in the range of [1,3] and a higher score is corresponding to a better example.

| Model | Response Quality | | | | | | Target-achievement | | |
|---|---|---|---|---|---|---|---|---|---|
| | **F1** | **BLEU-1** | **BLEU-2** | **DIST-1** | **DIST-2** | **Know. F1** | **SR@1** | **SR** | **Avg. Turns(↓)** |
| ChatGPT | 12.26 | 0.317 | 0.176 | **0.048** | **0.205** | 16.20 | 5.19 | 85.09 | 3.97 |
| RTCP (alpha = 0) | **45.39** | **0.542** | **0.402** | 0.036 | 0.109 | **70.35** | **9.21** | **86.78** | **3.80** |

Table 6: Automatic evaluation between RTCP and ChatGPT on the DuRecDial dataset.

| Model | Turn-level results | | | Dialog-level results | | |
|---|---|---|---|---|---|---|
| | **Fluency** | **Infor.** | **Proactivity** | **Satisfaction** | **Coherency** | **Kappa** |
| ChatGPT | 3.000 | 2.376 | 1.577 | 2.65 | 2.40 | 0.69 |
| RTCP | **2.981** | **2.246** | **1.676** | **2.55** | **2.45** | 0.74 |

Table 7: Human evaluation on the DuRecDial by RTCP and ChatGPT (**Infor.** stands for informativeness).

| | **DuRecDial** | **INSPIRED** |
|---|---|---|
| # of convs | 16.5K | 1001 |
| # of utterances | 255K | 35,811 |
| # of goals | 14 | 19 |
| # of topics | 646 | _ |
| # of domains | 5 | 1 |

Table 8: The detailed statistics of DuRecDial and IN-SPIRED datasets.

For a consistent evaluation, we carry out a set of detailed instructions to guide the annotators.

For fluency, we ask the annotators to check whether the generated responses are grammar-correct and understandable. The detailed instructions are as follows:

- (1): If the sentences are strictly grammar-incorrect and the annotators can not understand the meaning or the intention of the sentences (e.g.: "I 'm not sure if it is it , but I have n't seen it 's it 's a big").
- (2): The sentences might have some minor mistakes but they are still understandable.
- (3): The generated sentences are grammar-correct and the annotator can understand its meaning or intention (e.g.: "How about The Conjuring (2013) ?").

For informativeness, the annotators need to check if the generated responses contain topics or relevant information. The instructions are as follows.

- (1): The generated responses do not contain any useful information such as topics or related knowledge.
- (2): The generated responses contain either topics or related information.
- (3): The generated responses not only contain relevant topics but also provide additional related knowledge about the mentioned topics.

For proactivity, the annotators are required to check if the systems can use diverse intents (e.g.: chit-chat about a specific topic, ask clarifying questions, or recommend items) to interact with the users. The instructions are as follows:

- (1): The generated sentences are utilized to just passively respond to the user's last utterance (e.g.: "I haven't seen that movie."; "It is an interesting movie", etc.).
- (2): The goal of the generated responses is to obtain further information from the user. (e.g.: "What are your favorite singers ?"; "Do you like horror movies ?" etc.).
- (3): The generated responses are utilized to offer recommendations or to provide relevant information about a specific topic (e.g: "Speaking of Jacky Chung, he is also the Most Popular Asian Artist of Jade Solid Gold Best 10 Awards"; "Revolution is also very good. It's a song of Leehom Wang."; etc).

For satisfaction, we ask the annotators to evaluate how successfully the target-driven promotion task is fulfilled by considered methods. The detailed criteria are as follows:

- (1): The systems fail to recommend the targeted item. That means the targeted item never appear in the generated dialogues.
- (2): The systems successfully recommend the targeted item but they do not provide any convincing reasons to persuade the user. (for instance, they might intermediately suggest the designated item).
- (3): The systems not only recommend the targeted item successfully but they also convince the users with additional justifications.

For coherency, we ask the annotators to check if generated responses are relevant to their corresponding dialogue contexts and if these responses with the user's utterances together they form a meaningful conversation about the targeted item.

- (1): Major generated responses are not relevant to their historical contexts. (for instance, the models might frequently produce simple

responses such as "It is a good movie" or "What kinds of movie do you like ?" for every dialogue context).

- (2): Generated responses are relevant to their dialogue contexts. However, they together with the user's utterances do not form a meaningful conversation about the targeted item.
- (3): Dialogues that satisfy the two criteria.