# OpenReview forum: "Reinforced Target-driven Conversational Promotion"
_EMNLP/2023/Conference — EMNLP 2023 Main_

### Official Review · Reviewer_nCqR · 2023-08-01

**Soundness:** 4

**Excitement:**

4: Strong: This paper deepens the understanding of some phenomenon or lowers the barriers to an existing research direction.

**Paper Topic And Main Contributions:**

This paper explores a promising yet under-explored conversational task, namely conversational promotion. It aims to promote specified items through engaging conversations. The authors propose a Reinforced Target-driven Conversational Promotion (RTCP) framework to perform long short-term strategic planning to predict a dialogue action (a pair of goal and topic), then employ prefix-tuning to generate proper responses guided by the planned actions. Experiments on two datasets demonstrate the effectiveness of the proposed approach. Overall, this paper is well-written, with clear motivations and good experimental settings.

**Questions For The Authors:**

A. What does the task-specific token $T_{gen}$ refer to in Line 357? How important is it during prefix tuning?

B. In target-driven conversations, how to measure the average number of conversation turns? Are shorter conversations better?

**Reasons To Accept:**

1. This paper is well-motivated. Target-driven conversational promotion is an interesting yet challenging scenario, where how to conduct effective planning to promote target items proactively is a noteworthy research focus.
2. The proposed RTCP framework is reasonable and in line with the motivation. I really like the idea of making a balance between short-term planning and long-term planning.
3. Experiments are solid, with representative baseline models included and widely-used metrics for evaluation. Besides, it's good to see that the authors conduct additional analysis and show that RTCP can adapt to unseen conversation scenarios without whole re-training.

**Reasons To Reject:**

1. The term use of "dialogue actions" in this paper needs careful consideration. It might not be appropriate enough to define a dialogue action $a_t$ as a pair of goal $g_t$ and topic $p_t$. Generally, a dialogue topic outlines the subject or theme at a turn, while a dialogue action denotes the action or behavior performed by a speaker, e.g., asking for clarification, informing the user, making recommendations, etc. Actions and topics are two types of ingredients in the dialogue setting. Actually, I think the "goals" in the context of this work should be called "actions", though the authors follow the term use of UNIMIND (Deng et al., 2023). Nonetheless, I suggest the authors take a more appropriate term to denote "a pair of goal $g_t$ and topic $p_t$".

2. Some details of the proposed approach are not clear enough. (1) It lacks necessary descriptions of how $h_{s_{t+1}}$ is obtained when computing the reward $r_s$. (2) The two loss functions $L_g$ and $L_p$ in Lines 243-244 are not correct, since $P_{g}(\cdot)$ and $P_{p}(\cdot)$ are estimated probabilities, rather than ground-truth distributions. They should be in the form of $P(p^{*})\log P(p)$ when the authors use ground-truth distributions.

**Reproducibility:**

4: Could mostly reproduce the results, but there may be some variation because of sample variance or minor variations in their interpretation of the protocol or method.

**Reviewer Confidence:**

4: Quite sure. I tried to check the important points carefully. It's unlikely, though conceivable, that I missed something that should affect my ratings.

---

> ### Author Rebuttal · Authors · 2023-08-29
>
> Thank you for your thoughtful review and valuable feedback. Below we address your concerns.
>
> **Q1: The term use of "dialogue actions" in this paper needs careful consideration.**
>
> **A1:** We appreciate your suggestions and will utilize the suggested term “a pair of a goal $g_t$ and a topic $p_t$” in our modified version.
>
> **Q2: Some details of the proposed approach are not clear enough. (1) It lacks necessary descriptions of how $h_{s_{t+1}}$ is obtained when computing the reward $r_s$. (2) The two loss functions $L_g$ and $L_p$ in Lines 243-244 are not correct.**
>
> **A2:** Please let us provide some clarifications to address these two concerns. For (1), to compute the representation of the next state $h_{s_{t+1}}$, we feed the next state $s_{t+1}$ (which is a sequence) through a BERT model and utilize the output corresponding to the [CLS] token. Additionally, to train the binary classification model, we consider sequences of actions from the training set as positive instances and randomly replace some elements in the positive instances to produce negative ones. We will make these details more clear in the paper. For (2), we appreciate your thoughtful comment and will correct these loss functions in our modified version. Actually, our current equations such as the one in line 361, it focuses on the dimension in the probability vector corresponding to the groundtruth token $y_{t,j}^*$ , because other dimentions will multiply with zeros. But we will make the loss functions more clear following common practice.
>
> **Q3: What does the task-specific token $T_{gen}$ refer to in Line 357? How important is it during prefix tuning?**
>
> **A3:** $T_{gen}$ refers to task-specific soft tokens which are automatically learned to instruct the generation task. We got inspired from the work [1], which shows that adding such Task-Specific Prompt enhances model performace. Different from the action and topic prefixes $G_{t}$ and $P_{t}$ , we randomly initialize $T_{gen}$ and gradually optimize its parameters during training. To show the impact of $T_{gen}$, we additionally report the performance of RTCP wihout $T_{gen}$ on both the DuRecDial and INSPIRED datasets. As can be seen in Table 6 and Table 7 below, removing the task-specific prompt tokens lead to significant drops in the performance, which demonstrates that these prompt tokens are indeed crucial to the performance of RTCP.
>
> ```other
> Table 6: Ablation study on response generation for DuRecDial (t-test, p-value < 0.05).
> ```
>
> | **Model**                | **PPL**  | **F1**    | **Bleu-1** | **Bleu-2** | **Dist-1** | **Dist-2** | **Know.F1** |
> | ------------------------ | -------- | --------- | ---------- | ---------- | ---------- | ---------- | ----------- |
> | RTCP (alpha = 0)         | **3.69** | **45.39** | **0.542**  | **0.402**  | **0.036**  | **0.109**  | **70.35**   |
> | RTCP - t_gen (alpha = 0) | 5.10     | 36.38     | 0.482      | 0.326      | 0.031      | 0.104      | 46.86       |
>
> ```other
> Table 7: Ablation study on response generation for INSPIRED (t-test, p-value < 0.05).
> ```
>
> | **Model**                | **PPL**   | **F1**    | **Bleu-1** | **Bleu-2** | **Dist-1** | **Dist-2** | **Know.F1** |
> | ------------------------ | --------- | --------- | ---------- | ---------- | ---------- | ---------- | ----------- |
> | RTCP (alpha = 0)         | **18.64** | **15.39** | **0.371**  | **0.168**  | **0.091**  | **0.211**  | **10.21**   |
> | RTCP - t_gen (alpha = 0) | 33.78     | 13.25     | 0.346      | 0.152      | 0.063      | 0.154      | 4.41        |
>
> [1] Wang, Xiaolei, et al. "Towards unified conversational recommender systems via knowledge-enhanced prompt learning." *In KDD*. 2022.
>
> **Q4: In target-driven conversations, how to measure the average number of conversation turns? Are shorter conversations better?**
>
> **A4:** This is indeed a good question. In target-driven setting, we define the number of conversation turns as the number of turns needed to successfully recommend the target item. Shorter conversations does not mean they are better. A desired target-driven model needs to engage the users with relevant and interesting information to gain their attention before suggesting the target item. If a system constantly recommends the target items in the early turns of the conversations, it might make the users annoyed, which inherently hurts their experience. This is supported via experiment results reported in the paper. Specifically, in Table 2 in the paper, one can observe that some general pretrained language models namely GPT2 and BART achieve relatively good performance on the SR@1 metric. However, in Table 4 in the paper, the satisfaction score of GPT2 is significantly lower than other target-driven models such as RTCP and UNIMIND, which indicates that the annotators might not be so satisfied with responses generated by vanilla GPT2.

---

### Official Review · Reviewer_N7Uc · 2023-08-05

**Soundness:** 3

**Excitement:**

3: Ambivalent: It has merits (e.g., it reports state-of-the-art results, the idea is nice), but there are key weaknesses (e.g., it describes incremental work), and it can significantly benefit from another round of revision. However, I won't object to accepting it if my co-reviewers champion it.

**Paper Topic And Main Contributions:**

The paper introduces an approach to conversational recommendation by focusing on the strategic planning of nudging users towards accepting specific items. To address this, the authors propose the Reinforced Target-driven Conversational Promotion framework, which integrates short-term and long-term planning through a gating mechanism. This approach predicts dialogue actions using knowledge-integrated multi-head attention and also reinforces them through reinforcement learning rewards. Additionally, RTCP employs action-guided prefix tuning to generate relevant responses. Experimental results showcase the performance of RTCP over state-of-the-art models in terms of automatic metrics and human evaluations.


**Questions For The Authors:**

Not clear why parameters are learnt in line 354  shouldnt the values learnt from previous steps be directly used?


**Reasons To Accept:**

- The RTCP framework outperforms other baselines on the datasets considered.
- Ablation show the balanced gating mechanism and action-guided prefix tuning contribute to effective short-term and long-term planning and relevant response generation.


**Reasons To Reject:**

- The paper lacks clear explanation and detail regarding the metrics used for evaluation. Also not clear how much data was used in human eval.
- The performance results, especially in Table 1, show marginal differences from baselines.
- Not clear why parameters are learnt in line 354  shouldnt the values learnt from previous steps be directly used?


**Reproducibility:**

4: Could mostly reproduce the results, but there may be some variation because of sample variance or minor variations in their interpretation of the protocol or method.

**Reviewer Confidence:**

3: Pretty sure, but there's a chance I missed something. Although I have a good feel for this area in general, I did not carefully check the paper's details, e.g., the math, experimental design, or novelty.

---

> ### Author Rebuttal · Authors · 2023-08-29
>
> Thank you for your thoughtful review and valuable feedback. Below we address your concerns.
>
> **Q1: The paper lacks clear explanation and detail regarding the metrics used for evaluation. Also not clear how much data was used in human eval.**
>
> **A1:** We appreciate your suggestion and will additionally describe explanations regarding the evaluation metrics. In particular, for the response generation task, we utilize both automatic and human evaluation. For automatic evaluation, we use perplexity (**PPL**), word-level F1 (**F1**), **BLEU-N** (N=1,2), **Dist-N** (N =1,2) and Knowledge f1 (**Know. F1**).  Specifically, we use the perplexity (**PPL**) to quantify how uncertain the model is about its predicted responses. We use distinct (**Dist-N**) to evaluate the diversity of generated responses. **F1** and **BLEU-N** measure the similarity between the generated responses and ground truth sentences. The **Know. F1** shows the matching performance between generated knowledge and the ground truth knowledge triples. We will add more details in Appendix.
>
> For human study, we randomly sample 20 dialogues generated by RTCP and baseline models (shown in line 406). We then invite two annotators to score the responses in both turn-level and dialogue-level. The range of score is 1 to 3. The final performance is calculated using the average scores of all annotators. We measure the inter-annotator agreement by Fleiss’ Kappa. Human evaluation instruction details are described in Appendix A.6.
>
> **Q2: The performance results, especially in Table 1, show marginal differences from baselines.**
>
> **A2:** Thank you for pointing this out. We need to highlight that Table 1 (in paper) results are only for short-term planning. For maintaining smooth and coherent conversations, it is important for the agent to ensure not moving far away from the context. Also, the performance improvement are not marginal actually. For instance, in Table 1 (in paper) , compared to the best performing UNIMIND model, our RTCP achieve considerable improvement on both DuRecDial (+2.67 \% on Topic Accuracy,  + 2.75 \% on Joint Accuracy) and INSPIRED (+ 7.64 \% on Goal Accuracy). Additionally, we also ensure the statistical significance of the reported results via t-test with p-value < 0.05.
>
> **Q3: Not clear why parameters are learnt in line 354 shouldn't the values learnt from previous steps be directly used?**
>
> **A3:** Please let us provide some clarifications about our method here and we will make it more clear in the draft later. There are two separate components in our proposed RTCP framework namely a planning module and a response generation model. For the planning module, we utilize encoder models based on BERT to encode knowledge, previous planning sequence, and conversation context. For the response generation part, we adopt a GPT2 model as our main pretrained backbone. Regarding line 354 in the paper, to train our response generation module, we freeze the parameters of the GPT2 model and only learn parameters of the action-topic prefixes $G_{t}$, $P_{t}$ and task-specific prompt tokens $T_{gen}$. Hence, we cannot use the values learnt from the previous BERT model.

---

### Official Review · Reviewer_pkX6 · 2023-08-10

**Soundness:** 3

**Excitement:**

3: Ambivalent: It has merits (e.g., it reports state-of-the-art results, the idea is nice), but there are key weaknesses (e.g., it describes incremental work), and it can significantly benefit from another round of revision. However, I won't object to accepting it if my co-reviewers champion it.

**Paper Topic And Main Contributions:**

This work proposes an RL-based framework for conversational recommender systems emphasizing modeling the plan to make users accept the designated item. This topic is interesting in the IR community. This work focuses on a neglected problem in the CRS field.

**Questions For The Authors:**

See W1-W3.

**Reasons To Accept:**

S1. This work focused on a neglected problem in CRS, which may promote the application of CRS.

S2. Well-organized and well-written.

S3. Comprehensive comparison to the related works and PLMs.

**Reasons To Reject:**

W1. A related baseline is not compared. 'Unified Conversational Recommendation Policy Learning via Graph-based Reinforcement Learning' (UNICORN) is cited but not compared in the experiment. As far as I know, UNICORN also has an RL pipeline to control what attributes to ask, which items to recommend, and when to ask or recommend. Based on the paper's scope, it is also a competitive baseline to be considered.

W2. Evaluation criteria. BLEU-n and DIST-n may not be comprehensive as n only takes 1 and 2. It is recommended to set n as 3 and 4 for generative tasks. Besides, there are some advanced evaluation metrics apart from n-gram-based metrics.

W3. Generative PLMs are not enough. It is recommended to compare it with ChatGPT since it is free and available.

**Reproducibility:**

4: Could mostly reproduce the results, but there may be some variation because of sample variance or minor variations in their interpretation of the protocol or method.

**Reviewer Confidence:**

4: Quite sure. I tried to check the important points carefully. It's unlikely, though conceivable, that I missed something that should affect my ratings.

---

> ### Author Rebuttal · Authors · 2023-08-29
>
> We sincerely thank you for your time and valuable comments. We answer the questions as below.
>
> **Q1: A related baseline is not compared. 'Unified Conversational Recommendation Policy Learning via Graph-based Reinforcement Learning' (UNICORN) is cited but not compared in the experiment.**
>
> **A1:** We sincerely appreciate your suggestion. We indeed considered it. However, UNICORN is designed for scenarios in which system attempts to ask clarifying questions about attributes/properties to gradually search for an optimal candidate set of items. In contrast, our desired setting is to nudge users to accept a predefined target item, which is significantly different from UNICORN’s setting. Besides, UNICORN is a question-based CRS model that is trained with historical ratings of the users. Their benchmark datasets are recommendation datasets like Yelp and LastFM, which is rather different from our natural conversation dataset.  Moreover, UNICORN uses predefined templates to interact with the users which inherently can not handle arbitrary responses by the users.
>
> **Q2: Evaluation criteria. BLEU-n and DIST-n may not be comprehensive as n only takes 1 and 2. It is recommended to set n as 3 and 4 for generative tasks. Besides, there are some advanced evaluation metrics apart from n-gram-based metrics.**
>
> **A2:** Thank you for your suggestions about evaluation metric. Due to space limitation, we only reported BLEU-N, DIST-N (N=1,2). We additionally report the performance on BLEU-N, DIST-N (N=3,4) here, which will be added to the appendix. As can be seen in Table 1 and 2, compared to other existing methods,  RTCP achieved better performance on 3 out-of 4 and 2 out-of-4 metrics for DuRecDial and INSPIRED respectively, which shows a similar trend with the reported results in the paper. Indeed, there are also advanced evaluation metrics such as embedding based metrics (e.g. BERTScore, BARTScore etc). We will add the results and adjust the tables accordingly.
>
> ```other
> Table 1: Performance on DuRecDial (t-test, p-value < 0.05):
> ```
>
> | **Model**        | **BLEU-3** | **BLEU-4** | **DIST-3** | **DIST-4** |
> | ---------------- | ---------- | ---------- | ---------- | ---------- |
> | BART             | 0.198      | 0.167      | 0.162      | 0.222      |
> | DialogGPT        | 0.252      | 0.221      | 0.160      | 0.235      |
> | GPT2             | 0.277      | 0.242      | 0.182      | **0.26**   |
> | KERS             | 0.198      | 0.164      | 0.046      | 0.064      |
> | TCP              | 0.294      | 0.257      | 0.179      | 0.258      |
> | UNIMIND          | 0.310      | 0.273      | 0.176      | 0.254      |
> | RTCP (alpha = 0) | **0.343**  | **0.307**  | **0.184**  | 0.259      |
>
> ```other
> Table 2: Performance on INSPIRED (t-test, p-value < 0.05):
> ```
>
> | **Model**        | **BLEU-3** | **BLEU-4** | **DIST-3** | **DIST-4** |
> | ---------------- | ---------- | ---------- | ---------- | ---------- |
> | BART             | 0.057      | 0.039      | 0.288      | 0.386      |
> | DialogGPT        | 0.081      | 0.06       | 0.188      | 0.256      |
> | GPT2             | 0.083      | 0.059      | 0.341      | 0.445      |
> | KERS             | 0.032      | 0.021      | 0.037      | 0.043      |
> | TCP              | 0.087      | 0.063      | 0.359      | 0.46       |
> | UNIMIND          | 0.086      | 0.063      | **0.382**  | **0.486**  |
> | RTCP (alpha = 0) | **0.101**  | **0.077**  | 0.333      | 0.433      |
>
> **Q3: Generative PLMs are not enough. It is recommended to compare it with ChatGPT since it is free and available.**
>
> **A3:** We appreciate your suggestions and additionally report the performance of ChatGPT. First,  we show the generation performance of RTCP and ChatGPT on DuRecDial in Table 3.  We can observe that RTCP significantly achieves better performance on several metrics including BLEU-N (N=1,2,3,4), F1 and Know. F1. This is reasonable since RTCP is optimized to mimic responses in the corpus. However, ChatGPT performs better than our RTCP on DIST-N metrics (N=1,2,3,4). This can be attributed to the fact that ChatGPT have been pre-trained on a massive amount of texts. Therefore, it could generate more diverse responses. Second, we show the target-achievement results of RTCP and ChatGPT in Table 4. We can observe that our RTCP outperforms ChatGPT on all metrics. This can be attributed to the planning module of RTCP which could produce appropriate plans to direct the conversations towards the target item, while ChatGPT tends to passively answer user queries. This can be seen in Table 5 which shows the results of human evaluation with RTCP and ChatGPT on DuRecDial. Athough ChatGPT is trained with large corpus and with huge amount of parameters, our model still manages to outperform it in Proactiveness.
>
> ```other
> Table 3: Generation performance of RTCP and ChatGPT on DuRecDial.
> ```
>
> | **Model**        | **F1**    | **BLEU-1** | **BLEU-2** | **BLEU-3** | **BLEU-4** | **DIST-1** | **DIST-2** | **DIST-3** | **DIST-4** | **Know.F1** |
> | ---------------- | --------- | ---------- | ---------- | ---------- | ---------- | ---------- | ---------- | ---------- | ---------- | ----------- |
> | RTCP (alpha = 0) | **45.39** | **0.542**  | **0.402**  | **0.343**  | **0.307**  | 0.036      | 0.109      | 0.184      | 0.259      | **70.35**   |
> | ChatGPT          | 12.26     | 0.317      | 0.176      | 0.081      | 0.048      | **0.048**  | **0.205**  | **0.37**   | **0.49**   | 16.2        |
>
> ```other
> Table 4: Target-achievement Results of RTCP and ChatGPT on DuRecDial Dataset.
> ```
>
> | **Model**        | **SR@1** | **SR**    | **Avg_turn** |
> | ---------------- | -------- | --------- | ------------ |
> | RTCP (alpha = 0) | **9.21** | **86.78** | **3.80**     |
> | ChatGPT          | 5.19     | 85.09     | 3.97         |
>
> ```other
> Table 5: Human Evaluation on DuRecDial for RTCP and ChatGPT.
> ```
>
> | **Model**        | **Fluency** | **Informativeness**    | **Proactivity** | **Statisfaction**    | **Coherency** | **Kappa** |
> | ---------------- | -------- | --------- | ------------ | --------- | ------------ | ------------ |
> | RTCP  | 2.981 | 2.246 | **1.676**     | 2.55 | **2.45**     | **0.74**|
> | ChatGPT   | **3.000**     | **2.376**     | 1.577         | **2.65** | 2.40     | 0.69|

---

### Meta-Review · Area_Chair_GLRz · 2023-09-25

**Recommendation:** 4

**Metareview:**

The paper proposes a conversational nudging technique that allows the system to proactively plan and suggest items, improving the final outcome of item selection by the user. The paper has done the necessary evaluation and their method is sound.

---

### Decision · Program_Chairs · 2023-10-07

**Decision:**

Accept-Main

**Comment:**

The paper proposes a conversational nudging technique that allows the system to proactively plan and suggest items, improving the final outcome of item selection by the user. The paper has done the necessary evaluation and their method is sound.